# AMF Inoculation Alleviates Molybdenum Toxicity to Maize by Protecting Leaf Performance

**DOI:** 10.3390/jof9040479

**Published:** 2023-04-16

**Authors:** Mengge Zhang, Zhaoyong Shi, Shichuan Lu, Fayuan Wang

**Affiliations:** 1College of Agriculture, Henan University of Science and Technology, Luoyang 471023, China; 2Luoyang Key Laboratory of Symbiotic Microorganism and Green Development, Luoyang 471023, China; 3Henan Engineering Research Center of Human Settlements, Luoyang 471023, China; 4College of Environment and Safety Engineering, Qingdao University of Science and Technology, Qingdao 266042, China

**Keywords:** arbuscular mycorrhizal fungi, molybdenum, transport, allocation, photosynthesis

## Abstract

The use of arbuscular mycorrhizal fungi (AMF) is a vital strategy for enhancing the phytoremediation of heavy metals. However, the role of AMF under molybdenum (Mo) stress is elusive. A pot culture experiment was conducted to explore the effects of AMF (*Claroideoglomus etunicatum* and *Rhizophagus intraradices*) inoculation on the uptake and transport of Mo and the physiological growth of maize plants under different levels of Mo addition (0, 100, 1000, and 2000 mg/kg). AMF inoculation significantly increased the biomass of maize plants, and the mycorrhizal dependency reached 222% at the Mo addition level of 1000 mg/kg. Additionally, AMF inoculation could induce different growth allocation strategies in response to Mo stress. Inoculation significantly reduced Mo transport, and the active accumulation of Mo in the roots reached 80% after inoculation at the high Mo concentration of 2000 mg/kg. In addition to enhancing the net photosynthetic and pigment content, inoculation also increased the biomass by enhancing the uptake of nutrients, including P, K, Zn, and Cu, to resist Mo stress. In conclusion, *C. etunicatum* and *R. intraradices* were tolerant to the Mo stress and could alleviate the Mo-induced phytotoxicity by regulating the allocation of Mo in plants and improving photosynthetic leaf pigment contents and the uptake of nutrition. Compared with *C. etunicatum*, *R. intraradices* showed a stronger tolerance to Mo, which was manifested by a stronger inhibition of Mo transport and a higher uptake of nutrient elements. Accordingly, AMF show potential for the bioremediation of Mo-polluted soil.

## 1. Introduction

Molybdenum (Mo) is a trace element occurring in soil. It is an essential nutrient for most plants and animals and plays a key role in a variety of biochemical processes [1]. As a transition metal element, Mo has an atomic number of 42, a relative atomic mass of 95.95, and seven isotopes. It exists in soil in the form of soluble compound molybdate, sodium molybdate ammonium molybdate, insoluble compound molybdenum trioxide, molybdenum disulfide, and calcium molybdate. The content range of Mo in soil is generally 0.2~6 mg/kg, and the average content is approximately 2.3 mg/kg [2]. At an optimal level of Mo concentration, it is beneficial to plant photosynthesis and respiration and thus promotes plant growth and development. However, but in excess, Mo accumulation in crops may enter the food chain, causing potential risks to humans and animals [3]. Ferguson et al. (1943) first found Mo toxicity in ruminants due to the consumption of forages with a high Mo content [4]. Subsequent studies have shown that forages with Mo levels greater than 10 mg/kg generally cause diarrhea and other signs of toxicity, especially in ruminants [5]. Therefore, the accumulation of Mo in crops should be a concern.

The Mo content in plants is relatively low, generally lower than 1.0 mg/kg under normal conditions, and there is no clear definition of the toxicity threshold of Mo to plants [6]. However, chlorosis and yellowing occur in plants in environments with high concentrations of Mo in the soil [7]. A high Mo content in agricultural soils is mainly caused by human activities. Molybdenum mining activities have become more intensive in recent years due to the increasing demand for Mo in industrial products [5]. Mining and smelting inevitably cause Mo to outflow into the soil, resulting in the occurrence of excessive Mo in the soil. Luanchuan County is rich in mineral resources, with Mo reserves that rank first in Asia [8]. At present, the development of mineral resources will not only promote social and economic development but can also cause different degrees of damage to the ecological environment around the mining area [9]. Additionally, because the heavy metal pollution suffered by the soil is hidden, irreversible, and difficult to control, it will cause hidden dangers to the surrounding farmland for a long time. This issue has attracted the attention of scholars [10,11]. Wang et al. (2021) explored the Mo content of agricultural soil near a mining area and found that the soil molybdenum concentration reached 264 mg/kg [12]. Jia et al. (2015) investigated a farmland around a Mo mine in southeastern China and showed that the highest levels of molybdenum in the soil were 87 times higher than what is considered acceptable in typical Chinese agricultural areas [13]. In addition, the Mo concentration in alfalfa ranged from 48.4 to 142.0 mg/kg in an agricultural field surrounding the Luanchuan Mo mining site, which may present a potential health risk [12]. However, the mechanism of uptake and transport of molybdenum by crops and the toxicity threshold of molybdenum to crops under soil molybdenum pollution have not received sufficient attention.

Previous studies have suggested that symbiotic microbes are an eco-friendly and economical soil remediation approach that can mitigate plant stress caused by environmental stresses [14,15,16,17]. As one of the most widespread microorganisms, arbuscular mycorrhizal fungi (AMF) can form a close reciprocal symbiosis with more than 80% of land plants [18,19]. AMF can greatly increase the ability of plants to absorb nutrients through the hyphal network and can thus directly and indirectly influence physiological characteristics to improve plant growth [20,21]. In addition to improving the ability of plants to absorb mineral nutrients and trace metals, AMF can affect metal accumulation and transport in plants to reduce toxicity under heavy metal stress [22,23]. Studies have also found that AMF decreased plants’ uptake of heavy metals such as Zn, Cd, and Pb, helping the plants to grow better in heavy-metal-stressed soil [24,25,26,27,28]. However, little is known about how AMF effect plant growth and Mo absorption and transport in soil under stress from Mo.

This study aimed to examine the effect of different AMF on plants experiencing different concentrations of Mo stress, according to changes in the parameters related to growth and photosynthetic fluorescence properties. The absorption and transport of elements in a maize plant under Mo stress were investigated to identify the transport mechanisms of Mo under the influence of AMF. Specifically, we aimed to determine: (1) whether AMF can regulate the migration and distribution of Mo in maize plants; (2) whether AMF enhance the tolerance of maize plants to Mo stress and the physiological mechanisms; and (3) the response of plants to different AMF under Mo stress.

## 2. Materials and Methods

### 2.1. Preparation of Materials

The geographical location of the test site is 34°41′ N, 112°27′ E, which is located in the temperate continental monsoon climate zone. The experiments were performed in a pot culture house. The experiment was a complete factorial combination of two factors: Mo concentrations and AM fungal inoculations, with four levels of Mo (0, 100, 1000, and 2000 mg/kg) and three AM treatments (Non-AM/CK, AM1, and AM2), providing a total of 12 treatments, each one with four replicates. The determination of the added Mo levels was based on previous studies and the results of soil Mo content surveys in molybdenum mining areas [29,30]. (NH_4_)_2_MoO_4_ (analytical grade) was used to set the Mo level, and in order to eliminate differences in the N content caused by different (NH_4_)_2_MoO_4_ addition levels, NH_4_NO_3_ was used to balance N. Soil was procured from the agricultural field and sterilized in an oven at 121 °C for 60 min. The inoculums of AM1 and AM2 represent *Claroideoglomus etunicatum* (*C. etunicatum*) and *Rhizophagus intraradices* (*R. intraradices*), respectively. For AM inoculation treatments, 10% AM inoculum was mixed uniformly into each pot. To maintain uniformity, non-inoculation treatments were supplemented with the same amount of sterilized inoculum. The soil properties were as follows: pH (1:2.5, soil/water) 7.86, 13.5 g/kg total N, 1.12 g/kg total P, 11.49 g/kg total K, and 2.23 mg/kg total Mo.

Maize seeds (*Zea mays* L.) were sterilized with a solution of H_2_O_2_ (10%, *v*/*v*) for ten minutes and then cleaned with distilled water to eradicate trace chemicals. Thereafter, 1.2 kg of soil was placed into each pot (top diameter of 15.6 cm, bottom diameter of 11.0 cm, and height of 13.5 cm). Inactivated and activated AMF (*C. etunicatum* (BEG 168) and *R. intraradices* (BEG 141)) were added at a rate of 100 g/kg of AMF inoculum to form two treatment groups: CK (control) and AMF inoculation. The strain was preserved and purified in the ecological environment laboratory of the Henan University of Science and Technology. The AMF inoculant was propagated using maize grown in sterilized sand consisting of a mixture of spores, mycelia, sand, and root fragments and containing approximately 1000 spores per 100 g. Five seeds (Zheng Dan 958, purchased from Hebei Huafeng Seed Industry Development Co., Ltd. Shijiazhuang, China) of a uniform size were sown in each pot. The pots were then randomly placed in a plant growth chamber at approximately 25–28 °C. They were irrigated with deionized water every two days. Three months after the seeds were sown, all roots and shoots in each pot were separately sampled, washed with tap water and deionized water, and harvested for analysis.

### 2.2. Sample Analysis

Before the plantlets were harvested, three leaves per plant were darkened for 30 min, and an FMS-2 chlorophyll fluorometer and an M-PEA analyzer (from Hansatech Instruments Ltd., Pentney, King’s Lynn, UK) were used to determine the chlorophyll fast fluorescence kinetic parameters. The net photosynthetic rate (*Pn*) was determined using a portable photosynthesis measuring system (Ll-6800; LICOR, Lincoln, NE, USA) [31].

At harvest, the plants’ leaves, stems, and roots were harvested, and fresh leaves were taken to evaluate the leaf pigment content, including chlorophyll (chl) and carotenoid contents. These were extracted in a 95% ethanol solution, and the absorbance was then measured at 470, 649, and 665 nm. The amounts of the Chl a, Chl b, and carotenoids were calculated [31].

The plant tissues were dried (105 °C for 30 min and 70 °C for 48 h), and the dry weight was determined. The tissues were then ground for further analysis. Dried plant tissue samples were digested with perchloric acid and nitric acid (HClO_4_:HNO_3_ = 1:4, *v*/*v*). An inductively coupled plasma optical emission spectrometer (ICP-OES) (Perkin-Elmer 8300, Danbury, CT, USA) was used to determine the concentration of elements including Mo, Zn, Cu, and K [32].

The AMF colonization effects on the total biomass and Mo translocation factor were obtained using the following calculations [33].
Mycorrhizal dependency (%) = (DW_inoculated_ − DW_uninoculated_)/DW_uninoculated_(1)
Translocation factor (TF) = C_Mo shoots_/C_Mo roots_(2)
where DW is the plant total day weight, DW _inoculated_ is the inoculated plant day weight (inoculation with AMF), DW _uninoculated_ is the uninoculated plant day weight, C_Mo shoots_ represents the Mo concentration in the shoots, and C_Mo roots_ representsthe Mo concentration in the roots.

### 2.3. Statistical Analysis

The data were analyzed using SPSS 25.0 software. A one-way ANOVA was performed at the *p* < 0.05 level, and Duncan’s test was used to determine the significant differences between treatments. The figures were made using Origin 21.0. Using IBM SPSS Amos software, a structural equation model (SEM) was used to evaluate the relationships among AMF, Mo addition, and biomass.

## 3. Results

### 3.1. Effects of AMF Inoculation on Plant Biomass under Mo Stress

AMF inoculation significantly increased the biomass of maize plants under conditions of different Mo concentrations (Figure 1a). When the Mo addition level was the highest, inoculation with AM2 caused the shoot dry weight to reach a maximum of 10.32 g/pot. At a Mo supplemental level of 1000 mg/kg, the shoot and root biomass of the uninoculated plants was the lowest, with values of 3.06 and 1.21 g/pot, respectively. Mycorrhizal dependency clearly indicated a distinct increasing trend under stress concentrations of 0–1000 mg/kg (Figure 1b). Under the treatment of a Mo addition level of 1000 mg/mg, the mycorrhizal dependency of the AM1 and AM2 treatments was the greatest at 218.3% and 222%, respectively. The lowest mycorrhizal dependency occurred at the zero-addition level, with 106.68% under AM1 inoculation.

### 3.2. Mo Concentration in Maize under Different Levels of Mo Addition 

Mo supplementation significantly increased the concentration of Mo in maize plants (Figure 2). The highest Mo concentrations in the roots, stems, and leaves of the maize plants all occurred at the Mo addition level of 100 mg/kg. With the increase in the Mo supplemental level, the Mo concentration in the stems and leaves of the maize plants showed downward trends under all the inoculated and CK treatment conditions. When the addition level was 2000 mg/kg, the Mo concentrations in the stems and leaves reached the lowest level, with 40.86 and 100.56 μg/g, respectively. The concentration of Mo in stems and leaves was decreased by AMF inoculation and reached a significant level at 100 μg/g (Figure 2b,c). Moreover, the Mo concentration in the leaves of maize plants inoculated with AM2 was significantly lower than the Mo concentration in the leaves of maize plants inoculated with AM1 (Figure 2c). 

### 3.3. Effects of AMF on Mo Distribution in Different Organs

With the increase in the Mo addition level, the TF showed a decreasing trend, and at 2000 mg/kg addition level, the inoculation of AM2 reached the lowest level at 0.09 (Figure 3a). The TF of the inoculation treatment was significantly lower than that of the CK treatment except that the Mo supplemental level was 100 mg/kg. Furthermore, the TF of Mo in the plants inoculated with AM2 was lower than that of the plants inoculated with AM1 at each level of Mo addition. When the Mo addition was 0 mg/kg, the TF of the CK treatment was significantly higher than that of the inoculation treatment.

In the AMF-inoculated treatment, there was a larger percentage of Mo accumulated in the roots than in the non-inoculated treatment, with the root contributing more to Mo retention than the stem and leaf (Figure 3b). In addition, in the treatment without the addition of Mo, the largest proportion of Mo was accumulated in leaves at 70% without inoculation treatment, and the accumulation of Mo in the roots increased after inoculation. AMF inoculation also consistently increased the accumulation of Mo in roots and decreased the accumulation in leaves in the Mo-added treatments. After inoculation, the highest relative root accumulation of Mo reached 80% when the Mo addition level was 2000 mg/kg.

### 3.4. Effects of AMF on Photosynthetic Parameters of Maize under Different Degrees of Mo Stress

AMF inoculation significantly improved the net photosynthetic rate of maize plants at each Mo addition level (Figure 4). With the increase in the Mo addition level, the net photosynthetic rate under non-inoculation treatments showed a downward trend. The highest net photosynthetic rate of 18.53 μmol m^−2^s^−1^ occurred at the Mo addition level of 100 mg/kg with AM2 inoculation. The net photosynthetic rate of maize showed a result of AM2 > AM1 > CK when the Mo addition levels were 0, 100, and 1000 mg/kg. 

Compared to the non-inoculation treatments, AMF inoculation significantly increased the Chl a and carotenoid contents but decreased the Chl b content in maize plants (Table 1). With an increase in the Mo addition level, the Chl a and carotenoid contents showed downward trends under CK treatment, and this negative phenomenon was alleviated by AMF inoculation. When the Mo addition level was 100 mg/kg, the Chl a and carotenoid contents were the highest after inoculation with AM1: 2.64 and 0.6 mg/g FW, respectively. Under the same Mo concentration and inoculation conditions, the Chl b content was the lowest at 0.18 mg/g FW. A variance analysis showed that the AMF had significant effects on Chl a, Chl b, and carotenoids, among which the effect on Chl a was the most significant, with an *F* value of 64.27. The interactions between Mo and AMF have a significant effect on Chl a and Chl b.

The specific activity parameter of photosystem II can reflect the absorption, capture, and transmission of light energy by photosynthetic organs well. AMF improved the photochemical efficiency (Fv/Fm) of photosystem II at different Mo addition levels and reached a significant level at the Mo concentration of 100 mg/kg (Figure 5a). AMF inoculation improved the light-harvesting ability of maize plants under different Mo stresses, although it did not reach a significant level. The highest potential activity (Fv/Fo) of photosystem II occurred at the Mo addition level of 100 mg/kg with AM1 inoculation, and was significantly higher than the CK treatment with a Mo addition level of 1000 mg/kg.

The photosynthetic performance index (PIabs) is an important factor affecting the comprehensive photosynthetic performance of plant leaves. It reflects the effect of stress on plant photosynthetic structure and evaluates the degree of stress and the utilization rate of light energy. In addition to the Mo concentration of 2000 mg/kg, the PIabs of maize plants was significantly increased under other levels of molybdenum addition (Figure 5c). Among the three different inoculation treatments, the minimum PIabs appeared at the Mo addition level of 1000 mg/kg, and the maximum PIabs appeared in at a Mo addition level of 0 mg/kg with inoculation with AM1. For a Mo addition level of 0 mg/kg, the PIabs of maize plants inoculated with AM1 was significantly higher than that of the maize plants inoculated with AM2.

### 3.5. Uptake of Nutrient in Plants Organs

The uptake of nutrients by the plants was also calculated (Figure 6). Based on the one-way ANOVA results, AM inoculation had significant influences on the nutrient acquisition of plant tissues. Compared to the treatments without inoculation, inoculation with AM1 and AM2 significantly increased the uptake of Zn and Cu by the maize root, and the effects of AM1 and AM2 treatments on their absorption were consistent, with no significant difference (Figure 6a,b). However, the uptake of Zn and Cu by maize leaves inoculated with AM2 was significantly higher than the uptake of Zn and Cu by leaves inoculated with AM1 at 1000 and 2000 mg/kg levels of Mo addition. In most cases, inoculation increased the uptake of K in maize tissues, while the uptake of K in the stems of maize inoculation with AM1 was significantly lower than that in the CK treatment at the Mo addition level of 2000 mg/kg (Figure 6c). Compared with the non-inoculation treatments, inoculation significantly increased P uptake by maize plants at all levels of Mo addition (Figure 6d). The uptake of P by maize leaves inoculated with AM2 was significantly higher than that of leaves inoculated with AM1 at 0 and 2000 mg/kg levels of Mo addition. In general, AM inoculation significantly increased the uptake of nutrients by maize tissues.

The Mo concentration in the stems and leaves was significantly influenced by AM inoculation, Mo addition, and their interaction (Table 2), while the root Mo concentration was not significantly influenced by AMF. The TF was influenced significantly by AM inoculation, Mo addition, and their interaction. Consistently, the stem uptake of Zn was also influenced significantly by the three factors. The uptake of P, Cu, and K by roots was influenced significantly by AM inoculation but not by Mo addition or their interaction. By comparing the *F* values, the effect of Mo addition on Mo concentration and TF was more pronounced than that of AM inoculation. Regarding nutrient uptake, AM inoculation showed more obvious influences than Mo addition.

### 3.6. The Effect of Physiological Characteristics on Maize Biomass under AMF and Mo Stress

The direct and indirect effects of AM inoculation, Mo addition in *Pn*, chlorophyll, carotenoid, Mo TF, and nutrient accumulation on maize biomass were further revealed using a structural equation model (Figure 7). Mo addition had a negative effect on the TF and Zn uptake, while AM inoculation had a significant positive effect on Zn uptake (*r* = 0.60, *p* < 0.001). AMF had favorable effect on maize physiological characteristics, with the greatest effect on Chl a (*r* = 0.73, *p* < 0.001), while the addition of Mo had a negative effect on maize physiological characteristics and significantly inhibited the *Pn* (*r* = −0.34, *p* < 0.001). Chla and *Pn* had a direct positive effect on the biomass (*r* = 0.25, *p* < 0.01; *r* =0.75, *p* < 0.001). At the same time, Mo addition and AM inoculation both had significant positive effects on biomass, and the values of *r* were 0.26 and 0.41, respectively. 

## 4. Discussion

In the present study, a high level of Mo addition did not significantly inhibit the biomass of maize plants, indicating that the tolerance of maize plants are strong to Mo even at a concentration of 2000 mg/kg. This also supports previous findings that monocots are more tolerant to Mo [3]. Compared with the non-inoculated plants, AM-inoculated plants had a higher shoot and root biomass under different Mo addition levels, suggesting that AM plants have a stronger tolerance to Mo stress. The positive effects of AM inoculation on plant biomass were also reported in previous studies [34,35,36]. Our present results confirm that the AM fungi *C. etunicatum* and *R. intraradices* are tolerant to Mo stress and can help plants resist Mo toxicity. The mycorrhizal dependency index indicated the contribution of AMF to their host’s biomass. In this study, the mycorrhizal dependency index was the highest for inoculated AM1 or AM2 at the Mo addition level of 1000 mg/kg. This result was similar to the result of You’s study, which showed that the host plants are most dependent on mycorrhizal fungi under higher concentrations of heavy metal stress [35]. Previous studies have demonstrated that AMF are present in various metal-contaminated sites and can develop tolerance to excessive amounts of various trace elements [37,38,39,40]. Similar results were found in this study as the AMF species showed a good Mo tolerance.

Kovacs et al. (2015) [41] and Xu et al. (2018) [6] showed that the concentration of Mo in maize and soybean plants increased with increasing of levels of Mo addition. The difference is that our present results show that the root Mo concentration did not increase with the increase in the addition level. In addition, the Mo concentration in the stems and leaves decreased with the increase in the Mo addition level. This indicates that there is a defense mechanism to prevent excessive amounts of Mo from entering the plant. One possible mechanism is that under high Mo concentration, plants reduce Mo concentration in their tissues by “biological dilution”, which involves increasing the biomass to reduce the toxicity to plants [42]. On the other hand, it can be seen that corn plants have a certain tolerance to molybdenum. Adriano reported that when the molybdenum content of soybean, cotton, and carrot leaves reached 80, 1585, and 1800 mg/kg respectively, no abnormal plant growth was found [43]. This phenomenon may cause certain toxicity to animals. For example, Ferguson et al. (1938) suggested that animal consumption of high-molybdenum forages or drinking high-molybdenum water can cause diarrhea [44]. Regarding this study, in order to reduce this risk, we inoculated two strains to reduce the transport of molybdenum to plants. However, our limitation is that we did not harvest the grains of the maize plants. Therefore, the next step of this study is to measure the Mo concentration in maize grains and evaluate the retention effect of the strain on Mo. As for the effect of AMF on plants, many diverse results were presented for the reduction or increase in the uptake of heavy metals by plants. Liu et al. (2015) found that inoculation enhanced the concentration of Cd in the shoots and roots of *Solanum nigrum* when the Cd addition levels were 25 and 50 mg/kg [45]. Zhang et al. (2019) insisted that AMF reduced the heavy metal concentration in the shoots and roots of maize [46]. Similar to Zhang, Luo et al. (2017) also found that inoculation reduced the metal concentration in the roots, shoots, and grains of rice. However, most studies suggested that inoculation tends to immobilize the heavy metal in the mycorrhizosphere and decrease the HM’s concentration in the plant shoot [47,48,49]. This is consistent with our findings that inoculation with *C. etunicatum* and *R. intraradices* reduced the concentration of Mo in both the stems and leaves of maize (Figure 2b,c). Previous studies suggested that inoculation with AM fungi can enhance plant tolerance to heavy metals by affecting the distribution of heavy metals in host plants [50,51]. Our present work also confirms the result that inoculation with AMF inhibited the transport of Mo from the underground to aboveground (Figure 3a). It was demonstrated that AMF affects the bioavailability of Mo in the host plants and promotes Mo retention in the roots. The relative accumulation of Mo in plant tissues also showed that inoculation increased the accumulation of Mo in maize roots (Figure 3b). Particularly at a high Mo concentration of 2000 mg/kg, the active accumulation of Mo in roots reached 80% after inoculation. These results also prove the theory that AMF produce a large network of mycelium to hold heavy metals, retaining heavy metals in the mycorrhizal system to prevent the excessive movement of heavy metals and thereby inhibiting their transfer to the shoots [52,53,54,55].

In addition to their tendency to immobilize heavy metals in the roots, AMF also promote the plant uptake of nutrients. It is well-known that nutrient exchange is a major benefit between host plants and AMF [21]. During the symbiosis process, AMF can alleviate the oxidative stress caused by heavy metals by improving the plants’ absorption of mineral nutrients [56,57,58]. As an immobile mineral nutrient in soil, P plays a key role in the growth and development of higher plants. Some plants have evolved strategies to obtain P from the soil through two means of direct P uptake and indirect (mycorrhizal) pathways. When phosphorus near the root zone is depleted, plants usually choose the second pathway to absorb phosphorus, that is, symbiosis with mycorrhiza [59,60]. Studies have shown that the mycorrhizal auxiliary pathway can provide 80% of the inorganic phosphorus in plants [21]. Our results also achieved similar conclusions that inoculation significantly increased the uptake of phosphorus in maize tissues at the different levels of Mo addition. 

Consistent with the results of previous studies that AM fungi help to absorb K and other nutrients [61], this study showed that AMF also promoted the uptake of K and the micronutrients Cu and Zn under Mo stress. Studies have shown that Zn is the specific activation ion of chloroplast carbonic anhydrase, which is a key point in controlling photosynthesis and provides the substrate for CO_2_ fixation, thus promoting photosynthesis to a certain extent [62]. At the same time, Zn is a synthesis of chlorophyll precursor zinc porphyrin; thus, Zn is conducive to promoting chlorophyll synthesis, and chlorophyll plays a vital role in photosynthesis [63]. Therefore, it is speculated that the promotion of chlorophyll synthesis by Zn may also be one of the reasons for enhancing photosynthesis. In our current study, the structural equation model confirms the speculation (Figure 7) that the stress of Mo on plants causes the plants to tend towards increasing their photosynthesis and chlorophyll content by absorbing the nutrient element Zn, thereby enhancing the resistance of the plants to Mo stress.

Studies on the inhibition of plant photosynthesis by heavy metals have also been reported [64,65,66]. Whether CK or inoculation treatment was received, *Pn* generally decreased with the increase in Mo stress, indicating that the Mo toxicity inhibited the ability of maize to assimilate carbon [67]. However, inoculation with AMF significantly increased net photosynthetic rate (*Pn*) and alleviated the negative effects of Mo stress. Chlorophyll is an important component of chloroplasts and plays an important role in the photosynthesis of plants, and higher chlorophyll contents in plants are essential for maintaining normal photosynthesis processes under environmental stress [68,69]. AMF inoculation increased the content of photosynthetic pigments significantly, including Chl a and carotenoids, under Mo stress. Consistently, Santana et al. (2019) and Chaturvedi et al. (2018) also found a similar result in the studies on *Canavalia ensiformis* and *Solanum melongena L* [70,71]. This may be related to the inhibition of heavy metal transfer to the leaves by AMF [50]. PIabs is a performance index based on absorbing light energy, which can accurately reflect the state of plant photosynthetic apparatus. In this present study, compared with no-AMF, AMF inoculation improved the PIabs of plants. AMF may promote with the synthesis of photosynthetic pigments and thus impact photosynthetic processes. The structural equation model confirmed that the inoculation increased the biomass of plants, mainly by directly increasing the net photosynthetic rate and the chlorophyll content of the plants. In addition, inoculation reduced the negative effects of molybdenum transport on photosynthetic pigments and the net photosynthetic rate. Wu et al. (2019) also suggested that AMF inoculation alleviated the inhibition of heavy metal stress on photosynthesis of *Phragmites australis* and thus promoted growth [72]. This result revealed that mycorrhizal symbiosis is an effective strategy to improve plant tolerance to molybdenum.

## 5. Conclusions

In conclusion, inoculation with *C. etunicatum* and *R. intraradices* can enhance plant tolerance to Mo by regulating Mo allocation in host plants, showing that inoculation decreased Mo translocation from the roots to the shoots and increased the accumulation of Mo in maize roots. Inoculation increased the uptake of nutrient elements, which is an important factor in promoting the growth of maize plants. Additionally, a structural equation model analysis showed that the increase in the chlorophyll content and photosynthesis was the direct reason for the increase in biomass after inoculation with AMF. Compared with the fungi of *C. etunicatum*, *R. intraradices* showed a stronger tolerance to Mo, demonstrating a stronger inhibition of Mo transport and a higher uptake of nutrient elements. These initial findings suggest the mechanisms of AMF–plant symbioses under Mo stress and demonstrate that AMF may play a beneficial role in the ecological restoration of Mo-contaminated sites.

## Figures and Tables

**Figure 1 jof-09-00479-f001:**
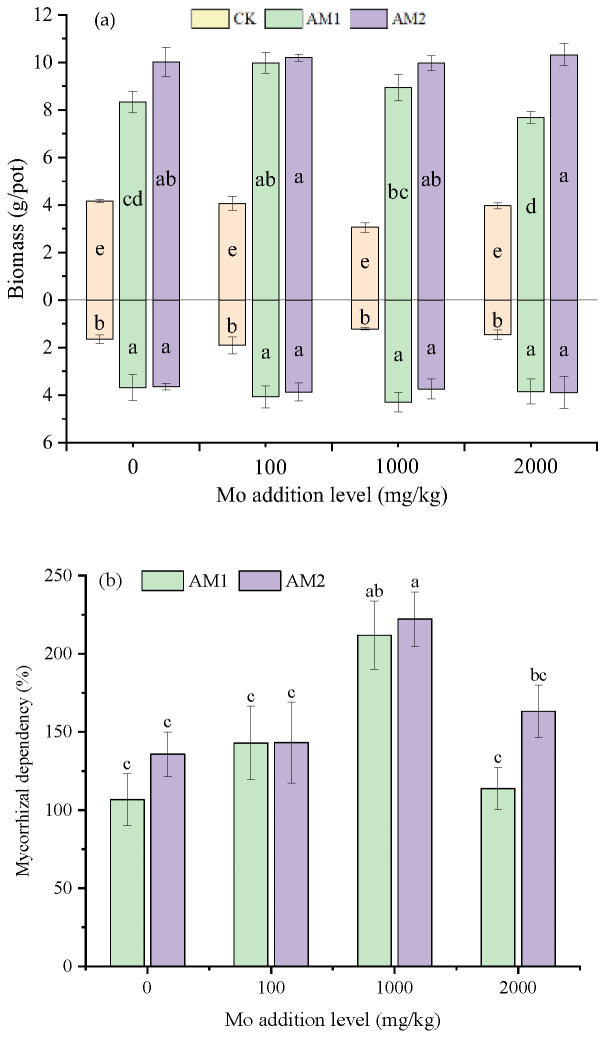
Shoot (above *X*-axis) and root (below *X*-axis) dry weight of maize plants and (**a**) mycorrhizal dependency (means ± SE, n = 4) (**b**) of maize at different levels of Mo addition. Different lowercase letters above the bars indicate significant differences under the different inoculation conditions among different levels of Mo addition, determined using a one-way ANOVA followed by Duncan’s multiple range test (*p* < 0.05).

**Figure 2 jof-09-00479-f002:**
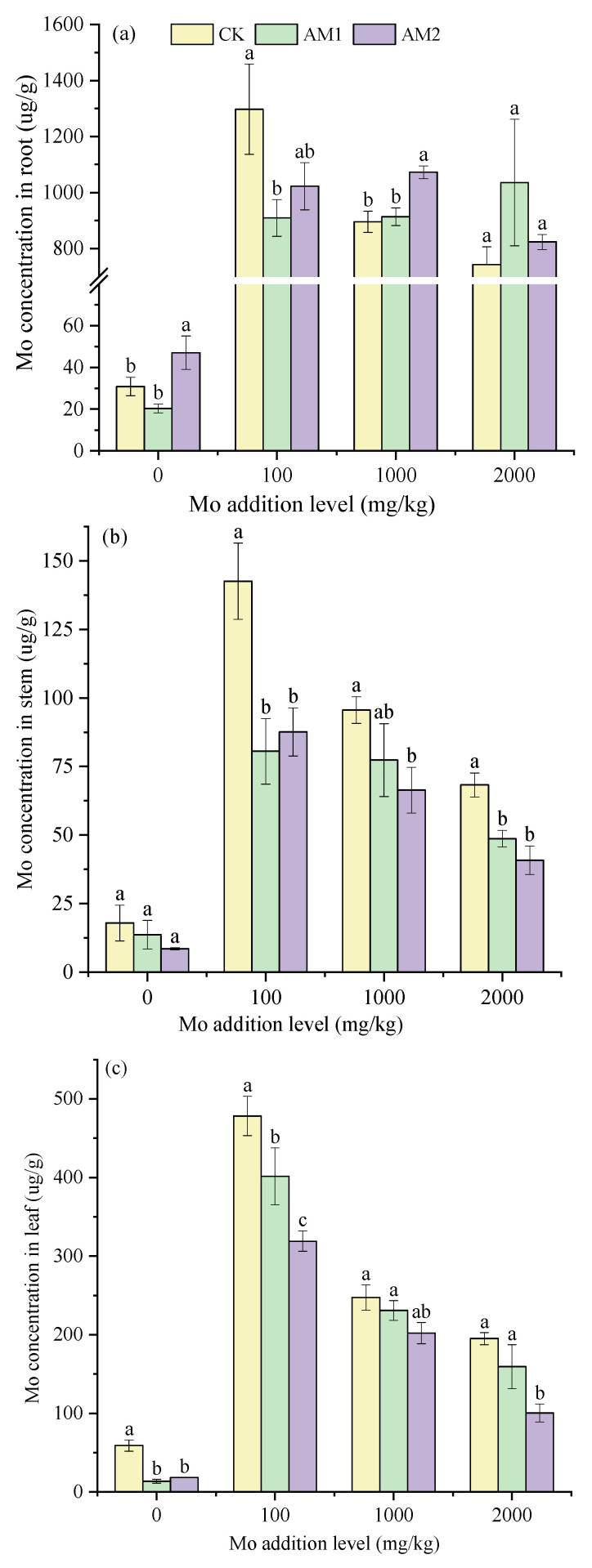
Mo concentration in root (**a**), stems, (**b**) and leaves (**c**) of maize under different levels of Mo addition. Different lowercase letters above the bars indicate significant differences under the different inoculation conditions at the same level of Mo addition, determined using a one-way ANOVA followed by Duncan’s multiple range test (*p* < 0.05).

**Figure 3 jof-09-00479-f003:**
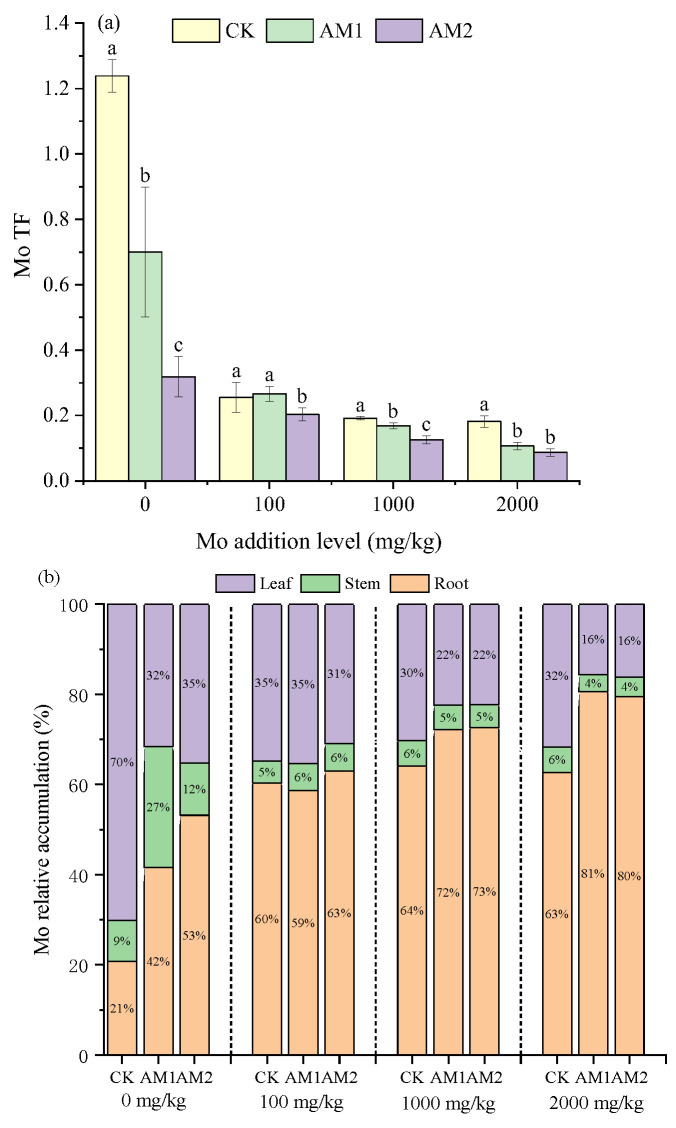
Translocation factor (**a**) and relative accumulation (**b**) under different levels of Mo addition in maize. Different lowercase letters above the bars indicate significant differences under the different inoculation conditions at the same level of Mo addition, determined using a one-way ANOVA fol-lowed by Duncan’s multiple range test (*p* < 0.05).

**Figure 4 jof-09-00479-f004:**
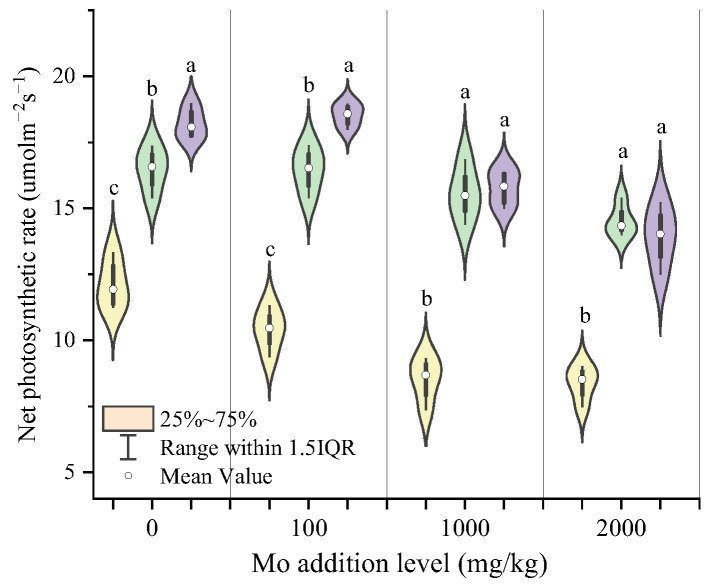
Net photosynthetic rate of maize at different Mo addition levels. Different lowercase letters above the bars indicate significant differences under the different in-oculation conditions at the same level of Mo addition, determined using a one-way ANOVA fol-lowed by Duncan’s multiple range test (*p* < 0.05).

**Figure 5 jof-09-00479-f005:**
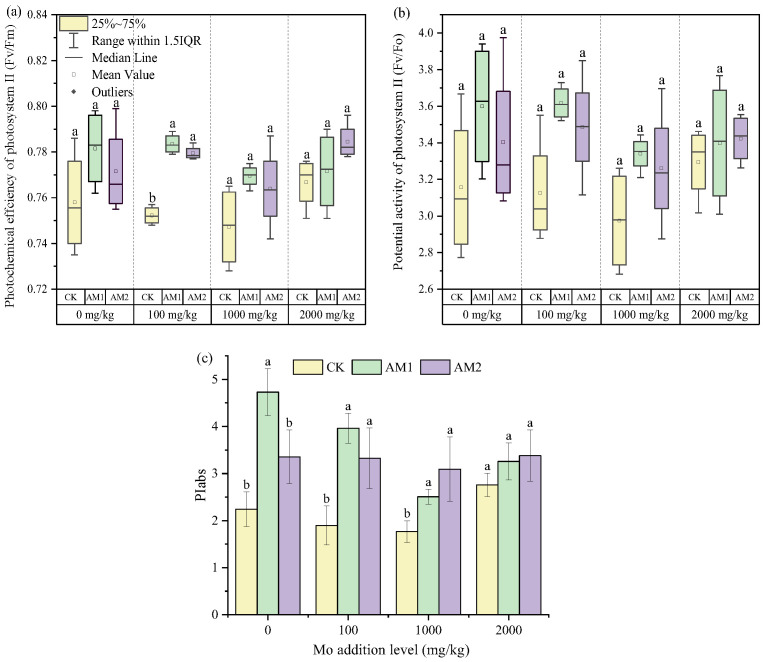
The specific activity parameter of photosystem II. photochemical efficiency (**a**), potential activity (**b**), and photosynthetic performance index (**c**). Different lowercase letters indicate significant differences under the different inoculation conditions at the same level of Mo addition, determined using a one-way ANOVA followed by Duncan’s multiple range test (*p* < 0.05).

**Figure 6 jof-09-00479-f006:**
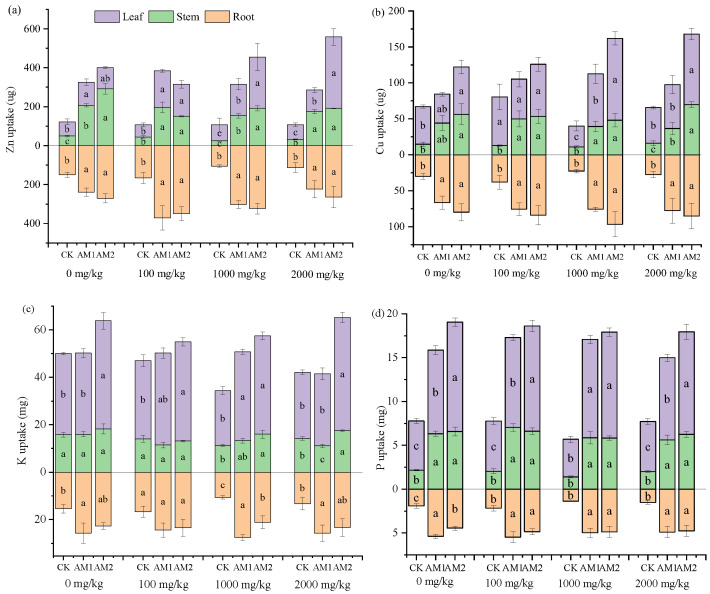
Shoot (above *X*-axis) and root (below *X*-axis) Zn (**a**), Cu (**b**), K (**c**), and P (**d**) uptake in maize plants at different Mo and AMF treatments. Vertical bars indicate standard deviations of the means (*n* = 4). Different lowercase letters indicate significant differences under the different inoculation conditions at the same Mo addition level, determined using a one-way ANOVA followed by Duncan’s multiple range test (*p* < 0.05).

**Figure 7 jof-09-00479-f007:**
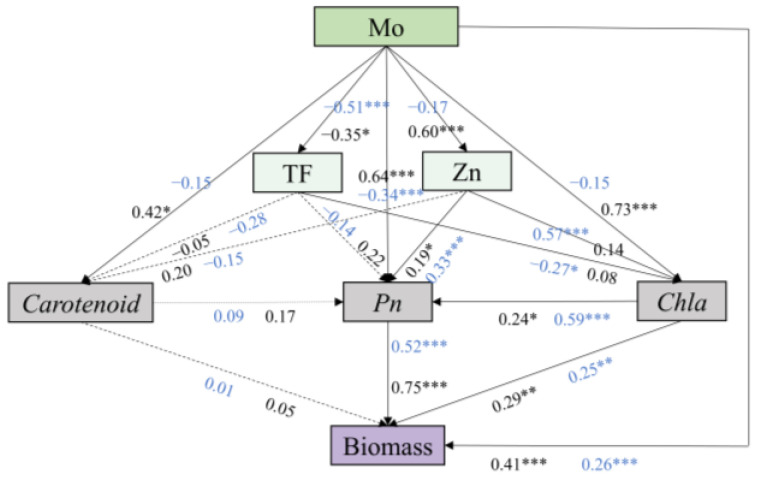
Structural equation model (SEM) analysis of the main pathways of AMF and Mo addition with maize physiological characteristics (Chla, *Pn*, and carotenoid), Mo TF, and nutrient uptake on maize biomass. Black and blue arrows represent AMF and Mo paths, respectively. Solid arrows represent significant positive or negative effects. Dashed arrows represent nonsignificant paths. Numbers near lines represent standardized path coefficients (*r*), * *p* < 0.05; ** *p* < 0.01; *** *p* < 0.001.

**Table 1 jof-09-00479-t001:** Chlorophyll content of maize leaf at different Mo addition and significance levels (*F* values) of Mo, AMF, and their interactions on measured variables by a two-way analysis of variance.

Photosynthetic Pigment		0 mg/kg	100 mg/kg	1000 mg/kg	2000 mg/kg	Mo	AMF	Mo × AMF
Chl a	CK	1.88 ± 0.08 b	1.67 ± 0.1 b	1.67 ± 0.16 b	1.63 ± 0.05 c			
	AM1	2.23 ± 0.1 a	2.64 ± 0.12 a	2.30 ± 0.11 a	2.14 ± 0.03 b	0.82 ns	64.27 **	2.56 *
	AM2	2.50 ± 0.08 a	2.45 ± 0.16 a	2.48 ± 0.12 a	2.61 ± 0.1 a			
Chl b	CK	0.23 ± 0.01 a	0.32 ± 0.01 a	0.28 ± 0.02 a	0.28 ± 0.02 a			
	AM1	0.20 ± 0.02 a	0.18 ± 0.01 b	0.23 ± 0.03 a	0.24 ± 0.01 b	3.17 *	19.65 **	3.32 *
	AM2	0.21 ± 0.01 a	0.20 ± 0.02 b	0.25 ± 0.01 a	0.20 ± 0.02 b			
Carotenoid	CK	0.41 ± 0.03 b	0.40 ± 0.06 a	0.35 ± 0.04 b	0.34 ± 0.01 c			
	AM1	0.46 ± 0.04 b	0.60 ± 0.06 a	0.53 ± 0.05 a	0.43 ± 0 b	1.38 ns	23.78 **	1.33 ns
	AM2	0.59 ± 0.02 a	0.56 ± 0.03 a	0.57 ± 0.04 a	0.58 ± 0.04 a			

Different lowercase significant differences under the different inoculation at the same Mo addition level using a one-way ANOVA followed by Duncan’s multiple range test. Significance levels: ns: non-significance; * *p* < 0.05; ** *p* < 0.01.

**Table 2 jof-09-00479-t002:** Significance levels (*F* values) of AMF inoculation and Mo addition, and their interactions on measured variables by a two-way analysis of variance.

	AMF	Mo	Mo × AMF
Root Mo conc.	0.08 ns	84.61 ***	2.98 **
Stem Mo conc.	15.97 ***	66.76 ***	2.57 *
Leaf Mo conc.	23.19 ***	230.12 ***	2.74 *
TF of Mo	19.44 ***	61.86 ***	11.13 ***
Root Zn uptake	29.66 ***	4.36 *	0.86 ns
Stem Zn uptake	196.23 ***	12.99 ***	6.19 ***
Leaf Zn uptake	9.71 ***	1.72 ns	2.13 ns
Root P uptake	71.30 ***	0.66 ns	0.33 ns
Stem P uptake	173.39 ***	3.10 *	0.63 ns
Leaf P uptake	199.22 ***	0.35 ns	2.55 *
Root Cu uptake	27.65 ***	0.23 ns	0.34 ns
Stem Cu uptake	29.03 ***	0.52 ns	0.67 ns
Leaf Cu uptake	16.25 ***	2.22 ns	3.67 **
Root K uptake	19.45 ***	0.23 ns	0.52 ns
Stem K uptake	8.12 **	6.23 **	2.07 ns
Leaf K uptake	53.55 ***	2.80 ns	4.59 **

Significance levels: ns non-significance; * *p* < 0.05; ** *p* < 0.01; *** *p* < 0.001.

## Data Availability

The data presented in this study are available on request from the corresponding authors. The data are not publicly available due to privacy.

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
