# Peer review of "AMF Inoculation Alleviates Molybdenum Toxicity to Maize by Protecting Leaf Performance"

_jof, 2023, doi:10.3390/jof9040479_

Round 1
Reviewer 1 Report
The current manuscript entitled “AMF inoculation alleviates molybdenum toxicity to maize by protecting leaf performance” by Zhang et al. aimed to investigate the effects of two arbuscular mycorrhizal fungi (AMF: Claroideoglomus etunicatum and Rhizophagus intraradices) inoculation on the alleviation of molybdenum toxicity in maize plants. The authors conducted a pot experiment with different levels of molybdenum and AMF inoculation treatments. The results showed that AMF shows potential for the bioremediation of Mo-polluted soil with improved growth of maize plants. Furthermore, AMF inoculation was found to alleviate molybdenum toxicity by reducing oxidative stress and increasing the activities of antioxidant enzymes in maize plants. After a careful reading, I found this experiment interesting and the topic is current to address Mo-contamination. The manuscript is suitable for publication in JoF after moderate changes. My specific comments are:
1. Introduction: Line 32: add elemental characteristics of molybdenum (atomic number, weight, isotope forms, composition on earth, soil soluble and insoluble forms, etc.).
2. Line 80: Add geocoordinates and climate data of the experimental site.
3. Correct the degree symbol (°) in the whole manuscript.
4. Accession numbers of organisms are missing.
5. From where the seeds were procured (source and variety name)?
6. References for plant analysis methods are missing.
7. Results and discussion: commendable.
8. Overall, the authors concluded that maize accumulated significant contents of Mo in its vegetative parts, however, what are the possible consequences of consuming such Mo-contaminated crops by both animals and humans? This is the major concern of this study and should be highlighted in the conclusion section. Also, provide possible limitations and future suggestions for extending this work.
Author Response
Dear reviewer:
On behalf of my co-authors, we thank you very much for giving us an opportunity to revise our manuscript, we appreciate you very much for your positive and constructive comments and suggestions on our manuscript. We have studied reviewer’ comments carefully and have made revision in the manuscript. And We have also answered the reviewers’ comments by point to point. Please find the following. We have tried our best to revise our manuscript according to the comments.
It is the following:
- Introduction: Line 32: add elemental characteristics of molybdenum (atomic number, weight, isotope forms, composition on earth, soil soluble and insoluble forms, etc.).
Answer: Accept. Thank the reviewer’s constructive suggestions. We have added the elemental characteristics of molybdenum in line 34-38:
“As a transition metal element, molybdenum has an atomic number of 42, a relative atomic mass of 95.95, and seven isotopes. It exists in soil in the form of soluble compound molybdate, sodium molybdate ammonium molybdate and insoluble compound molybdenum trioxide, molybdenum disulfide and calcium molybdate. The content range of Mo in soil is generally 0.2-6 mg/kg and the average content is about 2.3 mg/kg [2]”
van Gestel, C.A.M.; McGrath, S.P.; Smolders, E.; Ortiz, M.D.; Borgman, E.; Verweij, R.A.; Buekers, J.; Oorts, K. Effect of long-term equilibration on the toxicity of molybdenum to soil organisms. Environ. Pollut. 2012, 162: 1-7. DOI: 10.1016/j.envpol.2011.10.013
- Line 80: Add geocoordinates and climate data of the experimental site.
Answer: Accept. The geocoordinates and climate data of the experimental site has been added in the line 94-95. It was the following:
“The geographical location of the test site is 34° 41' N, 112° 27' E, which is located in the temperate continental monsoon climate zone.”
- Correct the degree symbol (°) in the whole manuscript.
Answer: Accepted. Thank the reviewer's checking carefully. We are sorry for our carelessness. It has been corrected. At the same time, we have checked our symbol through the whole manuscript.
- Accession numbers of organisms are missing.
Answer: Accepted. The accession numbers of organism and the source of AMF strains were added in line 126-132.
“Inactivated and activated AMF (C. etunicatum and R. intraradices) were added at a rate of 100 g/kg of AMF inoculum to form two treatment groups: CK (control) and AMF inoculation. The strain was preserved and purified in the ecological environment laboratory of Henan University of Science and Technology. The AMF inoculant was propagated using maize grown in sterilized sand, consisting of a mixture of spores, mycelia, sand, and root fragments and contained approximately 1000 spores per 100 g.
- From where the seeds were procured (source and variety name)?
Answer: the source and name of seeds were added in line 103. “Five maize seeds (Zheng Dan 958, purchased from Hebei Huafeng Seed Industry Development Co., Ltd) with uniform size were sown in each pot.”.
- References for plant analysis methods are missing.
Answer: we have added the references for plant analysis in references 31 and 32. There are the following:
- Dawood, M.; Cao, F.; Jahangir, M.M.; Zhang, G.; Wu, F. Alleviation of aluminum toxicity by hydrogen sulfide is related to elevated ATPase, and suppressed aluminum uptake and oxidative stress in barley. J. Hazard. Mater. 2012, 209: 121–128. DOI: 10.1016/j.jhazmat.2011.12.076
- Chaves, E.S.; dos Santos, E.J.; Araujo, R.G.O.; Oliveira, J.V.; Frescura, V.L.A.; Curtius, A.J. Metals and phosphorus deter-mination in vegetable seeds used in the production of biodiesel by ICP OES and ICP-MS. Microchem J. 2010, 96: 71–76. DOI: 10.1016/j.microc.2010.01.021
- Results and discussion: commendable.
Answer: Thank the reviewer’s recognition.
- Overall, the authors concluded that maize accumulated significant contents of Mo in its vegetative parts, however, what are the possible consequences of consuming such Mo-contaminated crops by both animals and humans? This is the major concern of this study and should be highlighted in the conclusion section. Also, provide possible limitations and future suggestions for extending this work.
Answer: Accepted. Indeed, our results indicated that maize was able to accumulate Mo in its vegetative parts. The reviewer is correct. The toxicity of Mo-contaminated plant to animals had been concentrated. For example, Ferguson et al. (1938) suggested that animal consumption of high molybdenum forage or drinking high molybdenum water can cause diarrhea. In term of this study, in order to reduce this risk, we inoculated two strains to reduce the transport of molybdenum to plants, but the limitation is that we did not harvest the grains of maize plants. Therefore, the next step of this study is to measure the molybdenum content in maize grains and evaluate the retention effect of the strain on Mo. It has been added in line 343-351.
“On the other hand, it can be seen that maize plants have a certain tolerance to Mo. Adriano (1986) also reported that when the Mo content of soybean, cotton and carrot leaves reached 80, 1585 and 1800mg/kg respectively, no abnormal plant growth was found. This phenomenon may cause certain toxicity to animals. For example, Ferguson et al. (1938) suggested that animal consumption of high molybdenum forage or drinking high molybdenum water can cause diarrhea. In term of this study, in order to reduce this risk, we inoculated two strains to reduce the transport of molybdenum to plants, but the limitation is that we did not harvest the grains of maize plants. Therefore, the next step of this study is to measure the Mo concentration in maize grains and evaluate the retention effect of the strain on Mo. Therefore, reducing the concentration of heavy metals in plants is an effective way to prevent the toxicity to animals. As for the effect of AMF on plants, many diversity results were presented for reduce or increase the uptake of heavy metals by plants.”
Adriana. Trace elements in the terrestrial environment. New York: Springer- Verlag Inc. 1986, 329.
Ferguson, W.S.; Lewis, A.H.; Watson, S.J. Action of molybdenum in nutrition of milking cattle. Nature, 1938, 141: 553.

Reviewer 2 Report
A simply experimental design and study so the results obtained quite straightforward; easy to handle and understandable for readers.
Introduction; a bit of background of Mo in typical Chinese agricultural areas or study site would be worth to add for description then will improve the story.
Materials preparation; how these 2 AMF strains obtained from? Locally , native sources or commercial?
Methodology; page 3 line 100; 3 months after seed sowing, the plants were harvested for analysis. How may of plants were harvested, how to selected the representative?
how AMF infection rate in maize for these 2 species after 3 months? Does infection rate related to Mycorrhizal dependency? AM 1 yielded 218% and AM2 yielded 222% dependency . So these also related to higher AMF infection? This may be interesting results to see and discuss.
Author Response
Dear reviewer:
On behalf of my co-authors, we appreciate you very much for your positive and constructive comments and suggestions on our manuscript. We have studied reviewer’ comments carefully and have made revision in the manuscript. And We have also answered the reviewers’ comments by point to point. Please find the following.
- Introduction; a bit of background of Mo in typical Chinese agricultural areas or study site would be worth to add for description then will improve the story.
Answer: Thank the reviewer’s constructive comments. In introduction, We have added the background of Mo and study site in line 55-61. It is the following:
“Luanchuan County is rich in mineral resources, with the Mo reserves ranking first in Asia (Zeng et al., 2013). At present, the development of mineral resources will not only promote social and economic development, but also cause different degrees of damage to the ecological environment around the mining area (Wang et al., 2018). What’s more, because the heavy metal pollution suffered by the soil is hidden, irreversible, and difficult to control, it will cause hidden dangers to the surrounding farmland for a long time, which has attracted the attention of scholars (Zhang et al., 2020; Han et al., 2019).”
Reference:
- Zeng, Q.D.; Liu, J.M.; Qin, K.Z.; Fan, H.R.; Chu, S.X.; Wang, Y.B.; Zhou, L.L. Types, characteristics, and time-space distri-bution of molybdenum deposits in China. Int. Geol. Rev. 2013, 55: 1311-1358. DOI: 10.1080/00206814.2013.774195
- Wang, J.; Wang, X.; Li, J.K.; Zhang, H.X.; Xia, Y.; Chen, C.; Shen, Z.G.; Chen, Y.H. Several newly discovered Mo-enriched plants with a focus on Macleaya cordata. Environ. Sci. Pollut. Res. 2018, 25: 26493-26503. DOI: 10.1007/s11356-018-2641-7
- Zhang, Q.C.; Wang, C.C. Natural and human factors affect the distribution of soil heavy metal pollution: A review. Water Air Soil Pollut. 2020, 231: 7. DOI: 10.1007/s11270-020-04728-2
- Han, Z.X.; Wan, D.J.; Tian, H.X.; He, W.X.; Wang, Z.Q.; Liu, Q. Pollution assessment of heavy metals in soils and plants around a molybdenum mine in central China. Pol. J. Environ. Stud. 2019, 28: 123-133. DOI: 10.15244/pjoes/83693
- Materials preparation; how these 2 AMF strains obtained from? Locally, native sources or commercial?
Answer: The source of AMF was added in line 126-132.
“Inactivated and activated AMF (C. etunicatum and R. intraradices) were added at a rate of 100 g/kg of AMF inoculum to form two treatment groups: CK (control) and AMF inoculation. The strain was preserved and purified in the ecological environment laboratory of Henan University of Science and Technology. The AMF inoculant was propagated using maize grown in sterilized sand, consisting of a mixture of spores, mycelia, sand, and root fragments and contained approximately 1000 spores per 100 g.”
- Methodology; page 3 line 100; 3 months after seed sowing, the plants were harvested for analysis. How may of plants were harvested, how to selected the representative?
Answer: We have improved the way of harvesting plant samples in Methodology in line 135-137. It is the following:
“Three months after seed sowing, all roots and shoots in each pot were separately sam-pled, washed with tap water and deionized water, and harvested for analysis.”
Because it was a pot experiment, the plant samples were all harvested, so no representative plant samples were selected. This sampling method can be found in Wang 's study.
Wang, F.Y.; Cheng, P.; Zhang, S.Q.; Zhang, S.W.; Sun, Y.H. Contribution of arbuscular mycorrhizal fungi and soil amendments to remediation of a heavy metal-contaminated soil using sweet sorghum. Pedosphere, 2022, 32(6): 844-855. DOI: 10.1016/j.pedsph.2022.06.011
- how AMF infection rate in maize for these 2 species after 3 months? Does infection rate related to Mycorrhizal dependency? AM 1 yielded 218% and AM2 yielded 222% dependency . So these also related to higher AMF infection? This may be interesting results to see and discuss.
Answer: Thanks to the experts who provided us with a meaningful direction, as expert believe that Mycorrhizal dependence indicates the contribution of AMF to the host biomass. Besides, mycorrhizal dependence is related to many factors, such as environment, plant species and so on. For this experiment, the infection frequency of the two strains was consistent with100 %. Therefore, different strains may be the main reason for the difference of mycorrhizal dependence, because except for different strains, other environmental factors are carried out under constant conditions.
Because of the consistent infection frequency of the two bacteria, we did not put this part of the data in the article. If the experts think it is necessary to put this data, we will make further modifications. The infection frequency algorithm is as follows: According to the number of mycorrhizal structures in each segment of root system, 0%, 10%, 20%, 30% ... 100%, the infection rate of each root segment was determined. The mycorrhizal colonization can be calculated according to the formula: Σ (0 × the number of root segments + 10% × the number of root segments + 20% × the number of root segments + … + 100% × the number of root segments)/observed total number of root segments.

Reviewer 3 Report
The presented manuscript focused on the effect of AMF inoculation on the physiological parameters of maize in molybdenum-contaminated soil.
Molybdenum (Mo) is an essential element for most organisms including plants, microbes and animals, where plays a key role in a variety of different biochemical processes. However, both deficient and excessive exposure can cause severe abnormalities in the metabolism of living organisms. Mycorrhizal fungi can enhance plant tolerance to biotic and abiotic stresses like the presence of Mo in soil. Arbuscular mycorrhizal fungi (AMF) establish symbiotic associations with many plant species and are used for the bioremediation of metal-polluted sites. AM can alter the concentration of heavy metals in plants by immobilizing heavy metals in the cell wall of intra- or extra-radicular hyphae, chelating the metals by secretion of several compounds such as glomalin, or by compartmentalization of metals in fungal cells. Therefore, numerous studies have been conducted in recent years to investigate the effect of AMF on many different physiological parameters like gas exchange, chlorophyll fluorescence, and pigment content in presence of different heavy metals in soil. From this point of view, the studies presented for evaluation are not very innovative, but they raise an important issue for global agriculture. The submitted manuscript is well-written and I do not find any serious errors in it, either methodological or in the deduction.
Some of my comments are presented below:
L132-133 and Figure 1: As the biomass was expressed in grams per pot, my question is whether all the plants in the analysed pots survived. The number of individuals will, with such an accepted unit, have a big impact on the result. If there was a die-off of individuals, I propose to introduce also a percentage of mortality.
Discussion:
From which the greater metal retention capacity of R. intraradices may derive. Was this strain isolated from a Mo-contaminated environment? It would be worth mentioning in the M&M from where the AMF strains were obtained for the study.
L313: “Solanum nigrum” should be in italics
L376: “P. australis” mentioned for the first time and therefore the full species name should be given in italics.
Author Response
Dear reviewer:
On behalf of my co-authors, we thank you very much for giving us an opportunity to revise our manuscript, we appreciate you very much for your positive and constructive comments and suggestions on our manuscript. We have studied reviewer’ comments carefully and have made revision in the manuscript. And We have also answered the reviewers’ comments by point to point. Please find the following.
L132-133 and Figure 1: As the biomass was expressed in grams per pot, my question is whether all the plants in the analysed pots survived. The number of individuals will, with such an accepted unit, have a big impact on the result. If there was a die-off of individuals, I propose to introduce also a percentage of mortality.
Answer: Thank the reviewer 's comments, which gives us a great idea for the next research. In this study, there was no individual death, so individual mortality was not introduced. The experts are right that the survival of all the plants in the pot has a great impact on the biomass which expressed in grams per pot, and this proposal is very useful for our ongoing research. Thanks again for the proposal of the reviewer.
Discussion:
From which the greater metal retention capacity of R. intraradices may derive. Was this strain isolated from a Mo-contaminated environment? It would be worth mentioning in the M&M from where the AMF strains were obtained for the study.
Answer: Accept. Thank the reviewer’s constructive comments. Judging from the results, R. intraradices showed stronger tolerance to Mo, which showed stronger inhibition of Mo transport and higher uptake of nutrient elements. As experts say, this lead to greater metal retention capacity of R.intraradices. This strain was not isolated from a Mo-contaminated environment. We have added the source of AMF strains in the material method in line 126-132. It is the following:
“Inactivated and activated AMF (C. etunicatum (BEG 168) and R. intraradices (BEG 141)) were added at a rate of 100 g/kg of AMF inoculum to form two treatment groups: CK (control) and AMF inoculation. The strains were preserved and purified in the ecological environment laboratory of Henan University of Science and Technology. The AMF inoculant was propagated using maize grown in sterilized sand, consisting of a mixture of spores, mycelia, sand, and root fragments and contained approximately 1000 spores per 100 g.
L313: “Solanum nigrum” should be in italics
Answer: Accepted. Thank the reviewer's checking carefully. We are sorry for our carelessness. It has been corrected. At the same time, we have checked our symbol through the whole manuscript.
L376: “P. australis” mentioned for the first time and therefore the full species name should be given in italics.
Answer: Accepted. Thank the reviewer's checking carefully. It has been corrected.
